# Preparation of High Thermo-Stability and Compactness Microencapsulated Phase Change Materials with Polyurea/Polyurethane/Polyamine Three-Composition Shells through Interfacial Polymerization

**DOI:** 10.3390/ma15072479

**Published:** 2022-03-27

**Authors:** Shaofeng Lu, Qiaoyi Wang, Hongjuan Zhou, Wenzhao Shi, Yongsheng Zhang, Yayi Huang

**Affiliations:** School of Textile Science and Engineering, Xi’an Polytechnic University, Xi’an 710048, China; wangqiaoyi123456@163.com (Q.W.); zhj18392029163@163.com (H.Z.); shiwenzhao@xup.edu.cn (W.S.); z18451399978@163.com (Y.Z.); ditty-23@163.com (Y.H.)

**Keywords:** microcapsules, phase change materials, composite shell, interfacial polymerization, polyurea/polyurethane/polyamine

## Abstract

In the preparation of microencapsulated phase change materials (MicroPCMs) with a three-composition shell through interfacial polymerization, the particle size, phase change behaviors, core contents, encapsulation efficiency morphology, thermal stability and chemical structure were investigated. The compactness of the MicroPCMs was analyzed through high-temperature drying and weighing. The effect of the core/shell ratio and stirring rate of the system was studied. The results indicated that the microcapsules thus-obtained possessed a spherical shape and high thermal stability and the surfaces were intact and compact. Furthermore, in the emulsification stage, the stirring speed had a significant influence on the microcapsules’ particle size, and smaller particles could be obtained under the higher stirring speed, and the distributions were more uniform in these cases. When the core/shell ratio was lower than 4, both the core content and the encapsulation efficiency was high. Additionally, when the core/shell ratio was higher than 4, the encapsulation efficiency was decreased significantly. The three-composition shell greatly increased the compactness of microcapsules, and when the core/shell ratio was adjusted to 3, the mass loss of the MicroPCMs was lower than 6% after drying at 120 °C for 1 h. After the microencapsulation, double exothermic peaks appeared on the crystallization curve of the MicroPCMs, the crystallization mechanism was changed from the heterogeneous nucleation to the homogeneous nucleation and the super cooling degree was enhanced.

## 1. Introduction

Microencapsulated phase change materials (MicroPCMs) have expanded the application and market prospects of phase change materials [1,2,3,4], which include textiles, dyeing and printing [5,6,7], building energy saving [8,9,10], microencapsulated phase change slurry [11,12,13] and composite foam [14,15,16].

Microcapsules with polyurea shells have attracted more and more attention because they do not contain formaldehyde [17]. However, there are still some problems to be solved in the microcapsules with polyurea shells. The traditional polyurea shell formed through the reaction between diisocyanate and diamine in the aqueous phase has the problem of high permeability, and the thus-prepared microcapsule has poor stability and compactness at higher temperatures [18]. The compactness of microcapsules at high temperatures is particularly important for their application in textiles and clothing, so as to avoid the leakage of core materials during high-temperature treatment, because high-temperature treatment is a necessary step for textile coating curing. Therefore, more and more attention has been paid to research into thermal stability and the compactness of the microencapsulated phase change materials with polyuria shells [19,20]. It has been reported that increasing the shell thickness and increasing the stirring speed is beneficial [21,22], but the effect is limited. The increase in shell thickness will lead to a decrease in the enthalpy of the microencapsulated phase change materials. This is because the existing literature mainly focuses on single component polyurea or polyurethane shells [23,24]. Yin et al. selected tetraethyl orthosilicate (TEOS) as functional shell-forming monomers to enhance the thermal stability and compactness of traditional polyurea MicroPCMs [25], and the result indicated that the smoothness and compactness of both polyuria-SiO_2_ and polyurethane-SiO_2_ microcapsules was enhanced slightly when compared with that without TEOS. Hong et al. [26] prepared polyurethane-TiO_2_ microcapsules through interfacial polymerization, and the result indicated that the addition of TiO_2_ enhances the thermal stability and mechanical strength of the prepared composite microcapsules. Therefore, the preparation of composite shell microcapsules has become a research hotspot and point of focus in recent years.

In their former work aimed at improving the thermal stability and compactness of single-shell microcapsules, the authors investigated the MicroPCMs with shells composed of polyurea and polyurethane [27,28]. The results indicated that the thermal stability and compactness of the MicroPCMs with the polyurea/polyurethane composite shells were superior to those of the MicroPCMs with the shells made of a single composition. However, the MicroPCMs thus-obtained still suffered the drawback of core materials leakage. Additionally, the presented work was conducted with the intention of creating new MicroPCMs with the shell composed of three compositions, which were, respectively, polyurea, polyurethane and polyamine. The polyurea was formed through the reaction of 2,4-toluene diisocyanate (TDI) and diethylenetriamine (DETA), the polyurethane was formed between the unreacted isocyanate group with polypropylene glycol 2000 (PPG2000) and the polyamine was formed through the reaction of DETA and epichlorohydrin. In this study, the work is devoted to the exploration of multi-component composite shells and their impact on the thermal stability and compactness of MicroPCMs so as to provide an effective method for interfacial polymerization to prepare microcapsules with high thermal stability. The obtained MicroPCMs were characterized by differential scanning calorimetry (DSC), thermogravimetric analysis (TGA), Fourier transform infrared (FTIR) spectroscopy and scanning electron microscopy (SEM). The influence of emulsion stirring speed and core/shell ratio on the core content, encapsulation efficiency, thermal stability, compactness and surface morphology of microcapsules was studied. The satisfactory thermal stability and compactness of MicroPCMs were obtained by means of the preparation of the shell with three compositions.

## 2. Materials and Methods

### 2.1. Materials

Butyl stearate with a purity of 99.00 wt%, used as a core material, was purchased from Sinophram Chemical Reagent Co. LTD (Shanghai, China). In addition, 2,4-toluene diisocyanate (TDI), diethylenetriamine (DETA), polypropylene glycol 2000 (PPG2000) and epichlorohydrin with a purity of 99.00 wt% were used as shell monomers. TDI and DETA react to form a polyurea shell, TDI and PPG2000 react to form a polyurethane shell and DETA and epichlorohydrin react to form a polyamine shell. TDI was supplied by Chengdu United Chemical & Pharmaceutical Co. LTD (Chengdu, China). DETA and PPG2000 were purchased from Sinophram Chemical Reagent Co. LTD (Shanghai, China). Styrene maleic anhydride copolymer (SMA), as the emulsifier, was obtained from Hercules Company (Wilmington, DE, USA). All chemicals in this study were used without further purification unless otherwise specified.

### 2.2. Preparation of MicroPCMs

Here, 30 g of butyl stearate, 5 g of TDI, 1 g of PPG2000 and 1 g of epichlorohydrin were thoroughly mixed at 30 °C and the formed system was used as the disperse phase. Then, 1.5 g of SMA was added to 200 mL of the distilled water and the solution was used as the continuous phase. The disperse phase was added to the continuous phase at 30 °C, and the uniform O/W type emulsion was obtained through stirring at the speed of 5000 rpm for 8 min. To initiate the interfacial polymerization, 20 mL of aqueous solution containing 2 g DETA was slowly added to the aforementioned emulsion under continuous stirring. In the preparation of the polyurea outer shell, the system was kept at 30 °C for 30 min, and the temperature was increased to 70 °C under the stirring for 3 h to promote the reaction between the isocyanate and PPG2000 in the formation of the polyurethane shell. In the formation of the polyamine shell, 20 mL of aqueous solution containing 2 g DETA was continually added to the system and then reacted for 3 h to promote the reaction between the epichlorohydrin and DETA in the formation of the polyamine shell. The thus-obtained microcapsule slurry was filtered, rinsed repeatedly using distilled water at 60 °C and then dried to obtain the microcapsule powders.

### 2.3. Characterization of MicroPCMs

#### 2.3.1. Particle Size Distribution

The particle size distribution of the MicroPCMs was measured using an MS2000 laser particle size analyzer from Malvern (Malvern Instruments Limited, Malvern, UK).

#### 2.3.2. The Phase Change Properties

The phase change performances of samples were tested using a TAQ100 differential scanning calorimeter (DSC) from TA Instrument Company (New Castle, DE, USA) and nitrogen was used as the protective atmosphere and the heating rate was at 3 °C min.

#### 2.3.3. Core Content and Encapsulation Efficiency

The actual core content (*C_a_*) of microcapsules is defined as the percentage of phase change enthalpy of microcapsules (Δ*H_m_*) and phase change enthalpy of pure phase change materials (butyl stearate) (Δ*H_PCM_*), which can be obtained through Equation (1).
(1)$$C_{a}=\left(\frac{{\Delta H}_{m}}{{\Delta H}_{\mathrm{PCM}}}\right)\times 100$$

The theoretical core content (*C_t_*) of microcapsules is defined as the weight of *PCM* (*W_PCM_*) as a percentage of the weight of *PCM* (*W_PCM_*) and the weight of all monomers (*W_m_*) forming microcapsules, as shown in Equation (2).
(2)$$C_{t}=\frac{W_{\mathrm{PCM}}}{W_{\mathrm{PCM}}{+W}_{m}}\times 100$$

The encapsulation efficiency (E) of microcapsules is defined as the percentage of the actual core content (*C_a_*) to the theoretical core content, as shown in Equation (3).
(3)$$E=\left(\frac{C_{a}}{C_{t}}\right)\times 100$$

#### 2.3.4. Compactness Analysis

A certain amount of microcapsule slurry was sprayed on the paper and then dried continuously at 120 °C for 1 h. The quality of microcapsules was measured every ten minutes, and then the change of the weight loss rates could be examined. The data were measured three times on average.

#### 2.3.5. Thermal Gravimetric Analysis

The thermal stabilities of the samples were investigated using a TGA2 thermogravimetric analyzer (TGA) with a heating rate of 10 °C/min under a nitrogen atmosphere.

#### 2.3.6. Surface Characterization

The surface morphology of the prepared microcapsule slurry and the powder were observed by using an OLYMPUS CX41-32RFL fluorescence microscope and a KYKY-2008B scanning electron microscope (SEM).

#### 2.3.7. Chemical Structure

The chemical structure of microcapsules was obtained by using a Nicolet 5700 Fourier transform infrared (FTIR) spectrophotometer.

## 3. Results and Discussion

### 3.1. Analysis of Particle Size and the Distribution

In the emulsification stages, the stirring speed was adjusted at 3000, 4000, 5000 and 6000 rpm, and the relative particle sizes and the distributions of the obtained MicroPCMs are shown in Figure 1. 

As can be seen from Figure 1, 80% of the particle diameters were smaller than 10, 8, 6 and 5 µm, and the average particle diameters were 9.13, 5.85, 4.41 and 3.69 µm when the stirring speed was, respectively, at 3000 rpm, 4000 rpm, 5000 rpm and 6000 rpm in the emulsification stages. This suggests that the smaller average particle sizes were highly related to the narrower particle size distributions. The lower stirring speed would lead to a wider particle size distribution and the higher stirring speed would cause a narrower particle size distribution. The particle sizes are highly depended on the stirring speed.

The satisfactory microcapsule particles can be obtained when the stirring speed is adjusted higher in the emulsification stage. The problem is that the system temperature can be increased quickly under high-speed stirring and so the side reaction between TDI and water would be accelerated, and this would affect the entire polymerization adversely. It was found that the satisfactory particle size and the stable system temperature would be obtained when the stirring speed was adjusted at around 5000 rpm.

### 3.2. Analysis of Core Content and Encapsulation Efficiency

The core/shell mass ratio (core/shell ratio) of MicroPCMs significantly affects the performance of the microcapsule. The higher core content would lead to the higher latent heat in the phase change process, and the performance of temperature regulation could, thus, be enhanced. It is obvious that, along with the increase in the core/shell ratio, the tightness of the wrapping would be decreased and so the thermal stability and compactness declined. The core content and encapsulation efficiency of the microcapsules prepared in different core/shell ratios are shown in Figure 2.

It can be seen from Figure 2a that the actual core content was very low under the high core/shell ratio. The reason is that the insufficient shell content led to the insufficient encapsulation and there was significant core content leakage. The shell could be well formed when the core/shell ratio was relatively low and, in these cases, the actual content of the core material was increased along with the increase in the core/shell ratio. The results indicated that the core content and the encapsulation efficiency were both satisfactory when the core/shell ratio was between 3 and 4. Further increasing the shell content could only increase the absolute weight of the microcapsules, and the core content was decreased. As shown in Figure 2b the encapsulation efficiency was higher at a low core/shell ratio, since the shell content was high, and the thickness of the formed polymeric shell was increased and the wrapping ability was also high. The encapsulation efficiency reached its maximum value of 94.5% when the core/shell ratio was 2, and it was decreased continuously when the core/shell ratio was higher than 2.

When the core/shell ratio was 3, the effect of the stirring speed on the core content was also studied. According to Figure 3, the stirring speed in the emulsification affected the core content of the microcapsules. The core content of the microcapsules gradually increased along with the increase in the stirring speed. The lower core content was related to a lower stirring speed in the emulsification stage. Along with the increase in the stirring speed, the core content was increased and the particle size became uniform under a high stirring speed, and the core material was well encapsulated. Therefore, a higher core content and excellent encapsulation efficiency can be obtained at a higher stirring speed in the emulsification stage. However, the core content was decreased slightly when the stirring speed surpassed 5000 rpm.

### 3.3. Analysis of Thermal Stability and Compactness

The thermal stability and compactness of MicroPCMs play a key role in their practical application. When microcapsule products were dried at higher temperature, the core material may leak. The higher mass loss of core materials was accompanied with the lower compactness and thermal stability. The microcapsule slurry with different core/shell ratios were sprayed on paper and then dried continuously at 120 °C for 1 h. The influence of the core/shell ratio on the compactness of the microcapsule is shown in Figure 4.

Figure 4 indicates that the core/shell ratio has an influence on the compactness of the microcapsule. Along with drying, the mass loss of MicroPCMs was continuously increased. The mass loss rate of MicroPCMs was higher at a higher core/shell ratio because the microcapsule shell thickness and compactness were decreased in this case. It should be noticed that the mass loss of MicroPCMs was only 6% after 1 h drying when the core/shell ratio was 3, and the mass loss rate of the MicroPCMs with a single polyurea shell was as high as 15.1% under the same condition [29], which indicates that the compactness of the microcapsule with the shell composed of three compositions was much improved.

At the same time, the influence of the emulsification stirring speed on the compactness of microcapsules was studied. When the core/shell ratio was adjusted to 3 and the emulsification stirring speeds were different, the mass loss of the prepared composite shell MicroPCMs dried at 120 °C for 1 h was affected as shown in Figure 5.

Figure 5 indicates that with the increase in the stirring speed, the mass loss rates of MicroPCMs decreased continuously after being dried for 1 h. According to the results, it can be seen that the compactness of the microcapsules was improved when the particle size of the microcapsules decreased. The reason is that the stability of the emulsion system was improved at a higher stirring speed, and so the formation of the shell was more uniform and the compactness was improved. When the stirring speed was 6000 rpm, the mass loss rate of the composite shell microcapsules treated at 120 °C for 1 h was only 5.6%, while the mass loss rate of polyurea microcapsules reinforced with tetraethyl orthosilicate under the same test conditions was as high as 8% [25]. This fully shows that the three-component composite shell can effectively improve the compactness of microcapsules.

In Figure 6, the thermal stability of butyl stearate and MicroPCMs with different core/shell ratios was evaluated using thermogravimetric analysis (TGA). The weight of butyl stearate and the prepared MicroPCMs decreased with the increase in temperature. When the core/shell ratio was 2:1, 3:1 and 4:1, the thermal resistant temperatures of MicroPCMs were 245, 234 and 226 °C, respectively. The prepared polyurea/polyurethane/polyamine three-composition shell microcapsules have excellent thermal stability. With the increase in the core/shell ratio, the thermal resistant temperature of MicroPCMs decreased continuously. This is because with the increase in the core material, the shell becomes thinner and the protection ability for the core material is weakened.

### 3.4. Analysis of Surface Morphology

When the stirring speed in the emulsification stage was 5000 and 6000 rpm, the optical microscope and SEM were employed to observe the surface morphology of the prepared MicroPCMs, and the results are shown in Figure 7 and Figure 8, respectively.

According to Figure 7, the prepared MicroPCM particles were uniform in size and had excellent dispersion stability, and there was no particle agglomeration. Furthermore, as shown in Figure 8, the prepared microcapsules were spherical, the shell was intact and compact, and the size was relatively uniform. 

When the stirring speed in the emulsification stage was at 6000 rpm, the particle size of the prepared MicroPCMs was about 3–4 µm, and when the speed was at 5000 rpm, their particle size was 4–5 µm. With the increase in the stirring speed, the average particle size of the microcapsules decreased gradually. Meanwhile, the surface of the prepared microcapsule particles became rough, which would be attributed to DETA penetration into the shell of the microcapsules and the reaction between DETA and epichlorohydrin, which would form the polyamine shell. The surface of polyuria/polyurethane double-shell microcapsules prepared without epichlorohydrin is very smooth [27]. Therefore, in order to make the reaction run smoothly, the reaction between DETA and epichlorohydrin could only occur in the low-density part of the microcapsule shell. Through the formation of the third polyamine shell, the weakness or the parts with poor compactness of the microcapsule shell were enhanced and improved, and the compactness and thermo-stability was also improved.

### 3.5. Analysis of Thermal Storage Performance

The DSC curves of both the pure butyl stearate and the relative MicroPCMs with the shell of three compositions are shown in Figure 9 for when the core/shell ratio was 3 and the stirring speed was 5000 rpm.

As shown in Figure 9 the melting curves of the pure phase change materials were similar to those of the relative MicroPCMs and possessed only one heating endothermic peak. The melting temperature (T_m_) and the melting enthalpy (ΔH_m_) of pure butyl stearate were 23.2 °C and 122.0 J·g^−1^, respectively. The melting temperature of the MicroPCMs was 23.0 °C, the melting enthalpy was 84.5 J·g^−1^ and the thermal storage performance was satisfactory. Therefore, the three-composition shell also increases the enthalpy of microcapsules because the melting enthalpies of polyurea and polyurea/polyurethane were 79.7 and 85 J·g^−1^, respectively, under the same conditions [27,29]. According to the crystallization curve of pure butyl stearate, there was only one exothermic peak, and the crystallization temperature (T_c_) and crystallization enthalpy (ΔH_c_) were 19.1 °C and 123.5 J·g^−1^, respectively. However, there were two exothermic peaks on the crystallization curve of the MicroPCMs. The T_c_ values were 20.7 and 14.1 °C, respectively, and the ΔH_c_ was 83.8 J·g^−1^. This double exothermic peak makes the exothermic behavior at a higher temperature occur in a wider range or at a lower temperature, leading to supercooled crystallization. These multiple crystallization phenomena were caused by the change in the crystallization mechanism. The peak α corresponded to the heterogeneous nucleation crystallization peak, and the peak β corresponded to the homogeneous nucleation crystallization peak [30].

In the investigation of the influence of microcapsule particle size on its crystallization behavior, the crystallization behavior under different stirring speeds in emulsification stages was studied. The result was shown in Figure 10 and the relative data are summarized in Table 1.

From Figure 10 and Table 1, it can be seen that the phase change temperature was basically unchanged when the stirring speed in the emulsification stage was increased from 3000 to 6000 rpm. Meanwhile, the area of peak α was decreased from 71.2 to 31.2% and the area of peak β was increased from 28.7 to 68.7%. These results indicated that the PCM in the microcapsules changed from the heterogeneous nucleation to the homogeneous nucleation along with the increase in the stirring speed in the emulsification stage. Furthermore, the super cooling degree of MicroPCMs was increased, since the microcapsule particle size was decreased, and the crystal nucleus number in each microcapsule was reduced and the seed number in the heterogeneous nucleation was reduced [31].

### 3.6. Chemical Structure Analysis

When the core/shell ratio was 3 and the stirring speed was 5000 rpm, the FTIR spectra of butyl stearate and the MicroPCMs were as shown in Figure 11. Typically, the characteristic absorption peaks of carbonyl stretching vibrations for butyl stearate are usually at 1739 cm^−1^. The absorption peaks at 2926 cm^−1^ and 2851 cm^−1^ are the C-H stretching vibrations. The stretching vibration peaks of O-H and N-H overlap at 3310 and 1655 cm^−1^, corresponding to the carbonyl vibration of urea. Additionally, 1103 cm^−1^ is the stretching vibration of the ether group, and 1379 cm^−1^ is the characteristic absorption peak of polyoxypropylene. The absorption peaks at 760 and 887 cm^−1^ are the characteristic absorption peaks of the epoxy group. These characteristic peaks can, significantly, be found on the FTIR spectrum of the MicroPCMs as well, which confirms that the MicroPCMs are composed of the PCM, polyurea, polyurethane and polyurethane.

## 4. Conclusions

MicroPCMs with the shell composed of polyurea, polyurethane and polyamine and with butyl stearate as the core material were prepared through interfacial polymerization. The stirring speeds in the emulsification stages have a significant influence on the particle size of the microcapsules. When the core/shell ratio was adjusted to 3–4 the core content was high, and the encapsulation efficiency was satisfactory. When the core/shell ratio was lower than 3, the encapsulation efficiency was higher than 92%. The DSC analysis results showed that the crystallization mechanism of the phase change materials was changed after the microencapsulation, and this yielded the super cooling phenomenon. Meanwhile, the super cooling phenomenon became more obvious, along with the decrease in the microcapsule particle sizes. The images obtained from the detection of the optical microscope and scanning electron microscopy showed that the microcapsules with the shell composed of three compositions were spherical in shape and had a uniform size distribution. The shell was intact and compact, the surface was slightly rough and no agglomeration was found. The stirring speed in the emulsification stage and the core/shell ratio of the microcapsules had an important influence on the thermal stability.

## Figures and Tables

**Figure 1 materials-15-02479-f001:**
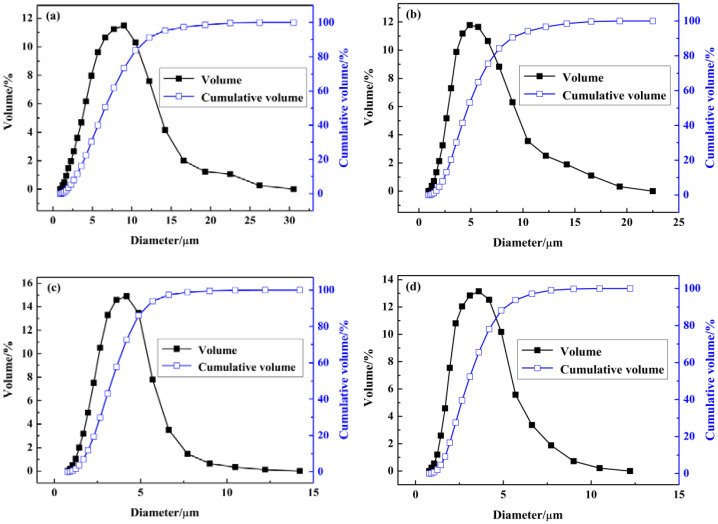
The influence of emulsification stirring speed on the particle size and distribution of the prepared MicroPCMs: (**a**) 3000 rpm, (**b**) 4000 rpm, (**c**) 5000 rpm, (**d**) 6000 rpm.

**Figure 2 materials-15-02479-f002:**
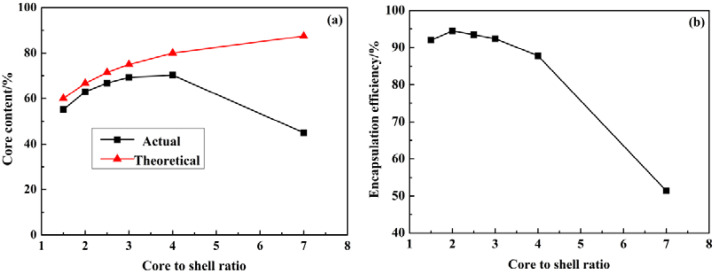
Influence of core/shell ratio on (**a**) core content and (**b**) encapsulation efficiency of MicroPCMs.

**Figure 3 materials-15-02479-f003:**
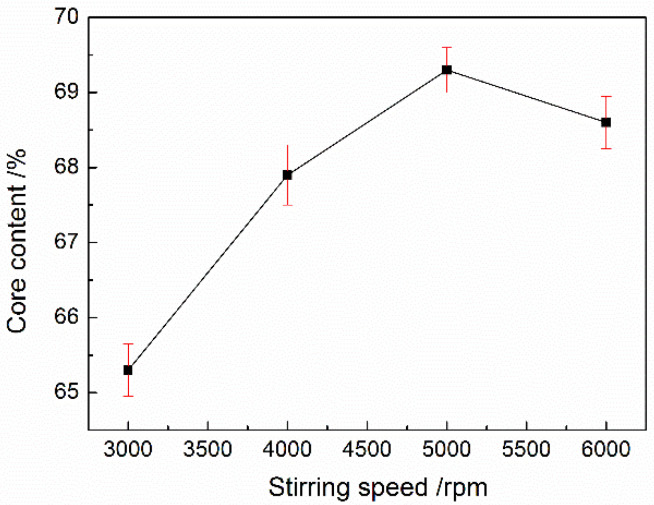
The influence of the stirring speed in emulsification on MicroPCMs’ core content.

**Figure 4 materials-15-02479-f004:**
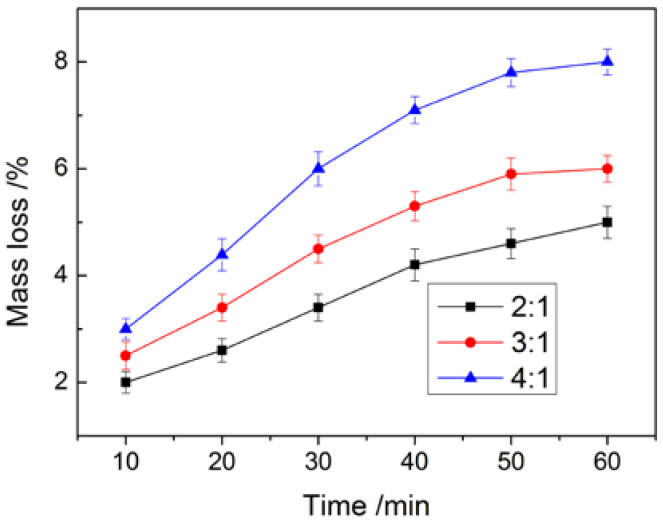
The influence of core/shell ratio on compactness of MicroPCMs.

**Figure 5 materials-15-02479-f005:**
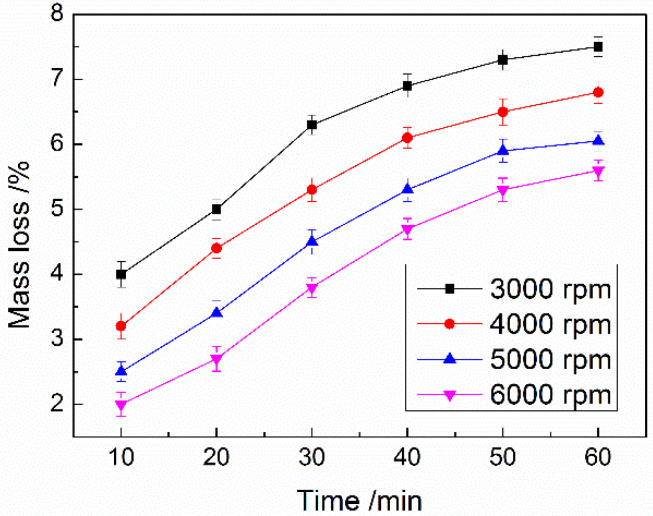
The influence of the stirring speed in the emulsification stages on compactness of MicroPCMs.

**Figure 6 materials-15-02479-f006:**
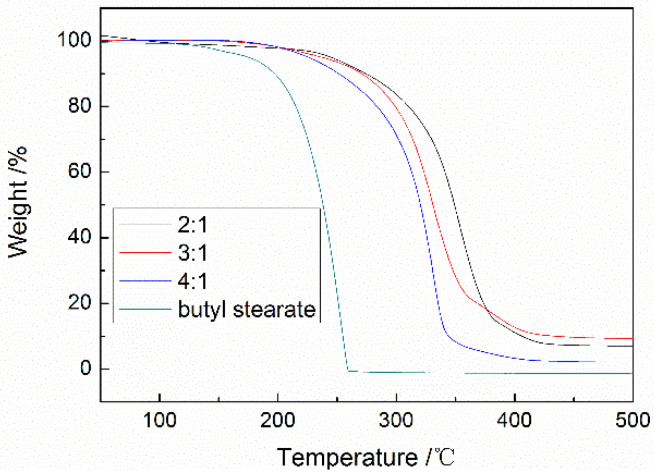
TGA curves of MicroPCMs with different core/shell ratio.

**Figure 7 materials-15-02479-f007:**
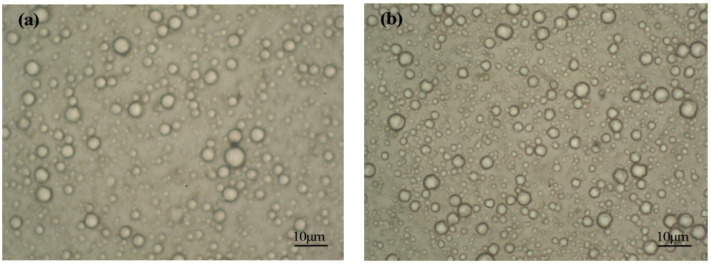
The optical microscope morphology of the prepared MicroPCMs at the emulsification stirring speed of (**a**) 5000 rpm and (**b**) 6000 rpm.

**Figure 8 materials-15-02479-f008:**
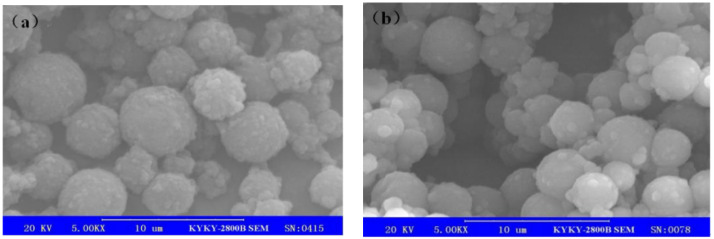
SEM morphology of the prepared MicroPCMs at the emulsification stirring speed of (**a**) 5000 rpm and (**b**) 6000 rpm.

**Figure 9 materials-15-02479-f009:**
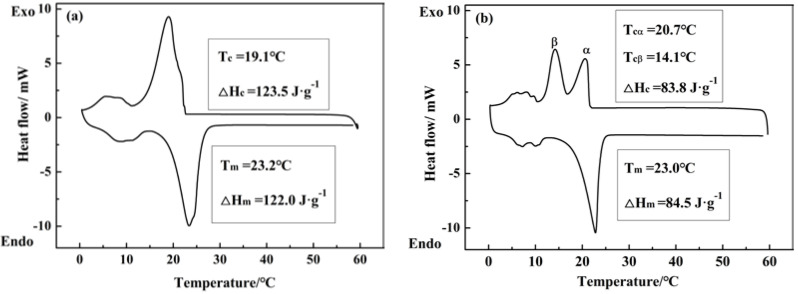
DSC curve of (**a**) butyl stearate and (**b**) MicroPCMs.

**Figure 10 materials-15-02479-f010:**
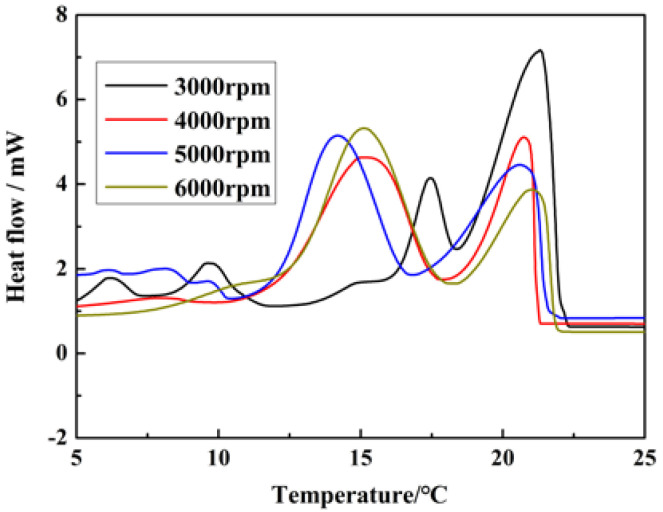
The influence of emulsification stirring speed on DSC crystallization curve of MicroPCMs.

**Figure 11 materials-15-02479-f011:**
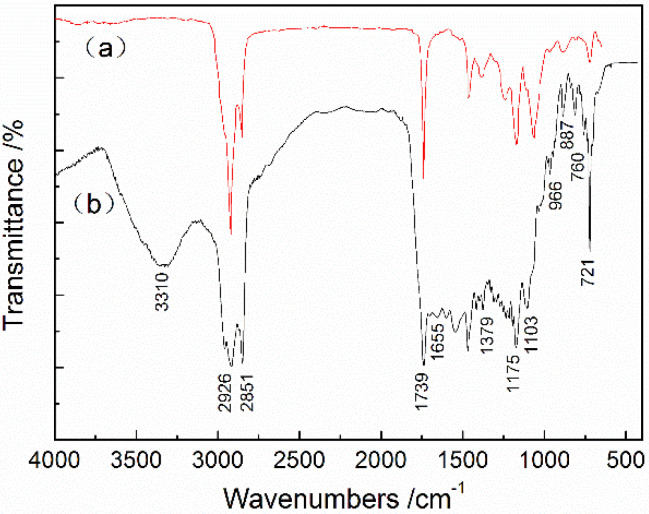
FTIR spectra of (**a**) butyl stearate and (**b**) MicroPCMs.

**Table 1 materials-15-02479-t001:** The crystallinity properties of the prepared MicroPCMs at different emulsification speeds.

Emu31lsification Speed/rpm	T_c_				ΔH_c_/J·g^−1^
	T_cα_/°C	A_α_ ^1^/%	T_cβ_/°C	A_β_/%	
3000	21.3	71.2	17.5	28.7	80.2
4000	20.8	45.1	15.4	54.9	83.1
5000	20.7	44.7	14.1	55.3	83.8
6000	21.2	31.2	15.3	68.7	84.1

^1^ A is the area percentage of peak α or β.

## Data Availability

Data sharing is not applicable to this article.

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
