# Peer review of "Preparation of High Thermo-Stability and Compactness Microencapsulated Phase Change Materials with Polyurea/Polyurethane/Polyamine Three-Composition Shells through Interfacial Polymerization"

_materials, 2022, doi:10.3390/ma15072479_

Round 1

Reviewer 1 Report

The authors present preparation of MicroPCMs containing polyurea, polyurethane and polyamine. The characterization and physical properties of the prepared MicroPCMs were studied by thermos gravimetric analyzer, differential scanning calorimeter and laser particle size analyzer. It was found that the average particle sizes decreased by increasing emulsification stirring speed. The encapsulation efficiency and the compactness of the MicroPCMs significantly decreased at high core to shell ratio. The DSC measurements showed satisfactory thermal storage properties. The article is well-written and worth to accept for publication but I have some remarks/questions.

The manuscript contains some typos, such as: in the last sentence of 2.2 section “60 oCand”, space missing; in 2.3 section DHPCM and WPCM, the PCM should be at subscript position; in the explanation of Fig 4, “In should be noticed” should be corrected It should be noticed etc.

In Fig 6 the thermal stability of butyl stearate is not shown.

I miss the deviation bars in some Figs, especially in Fig 3. Is the decreasing in the core content at 6000 rpm real or just a measurement error?

If the optimal value was 5000 rpm for the core content, what were the optimal rpm values for the other MicroPCMs with different core/shell ratios to obtain the highest core content %?

Reviewer 2 Report

The paper "Preparation of high stability Microencapsulated Phase Change Materials with polyurea/polyurethane/polyamine Three-Composition Shell through Interfacial Polymerization" is suitable for publication in Materials, as long as the authors can detail and improve the paper on some issues, such as:
(1) The title of the paper is not adequate, the authors should make it more objective and clear.
(2) There are yellow markings that should not exist, and the citation standard is different from that adopted in this journal.
(3) At the end of the introduction, the authors should more clearly show the objectives and innovation/motivation of this research to the readers.
(4) Item 2.1 (description of materials) should be better detailed and justified.
(5) The methodological description section is very incomplete, the authors should dedicate themselves to improving this topic.
(6) Some discussions should be better detailed in terms of comparison with works from other international literature.

Reviewer 3 Report

  • Authors are kindly requested to use line numbering in future manuscripts.
  • Paragraph 2.2: Authors should mention the role of the ingredients (e.g., emulsifier, phase change material, role of epichlorohydrin?) wherever is not given.
  • How is the polyamine shell formed? Does any kind of reaction or interaction take place? The term “polyamine” refers to DETA or some polymerized species?
  • The formation of the shell compounds must be verified by some characterization technique (e.g., FTIR, NMR, etc.), otherwise appropriate bibliographic sources should be referred.
  • Page 3, equation (3): The “Ca/Ct” ratio should be multiplied by 100
  • Page 7, 2nd paragraph: By reading the comments on Figure 6, someone expects to see the thermogram of neat butyl stearate, in addition to the thermograms of different MicroPCMs. Is the thermogram of butyl stearate missing?
  • Figure 7: Image 7b overlaps Image 7a and the scale bar disappeared.
  • Figure 9: To which sample does the DSC curve of Figure 9b correspond?

Round 2

Reviewer 2 Report

All corrections have been made.

Reviewer 3 Report

Thanks to the authors for their response to the comments and congratulations on their work